# Doloris: Dual Conditional Diffusion Implicit Bridges with Sparsity Masking Strategy for Unpaired Single-Cell Perturbation Estimation

**Changxi Chi**[1,2], **Jun Xia**[3], **Yufei Huang**[1,2], **Zhuoli Ouyang**[5], **Cheng Tan**[6],
**Yunfan Liu**[2], **Jingbo Zhou**[2], **Chang Yu**[2], **Liangyu Yuan**[7], **Siyuan Li**[2], **Zelin Zang**[4*]
and **Stan Z. Li**[2*]

[1]Zhejiang University    [2]Westlake University
[3]The Hong Kong University of Science and Technology (Guangzhou)
[4]Centre for Artificial Intelligence and Robotics Hong Kong Institute of Science & Innovation, Chinese Academy of Sciences
[5]Southern University of Science and Technology    [6]Shanghai AI Laboratory
[7]Shanghai Jiao Tong University
`{chichangxi, xiajun, zangzelin}@westlake.edu.cn`

## Abstract

Estimating single-cell responses across various perturbations facilitates the identification of key genes and enhances drug screening, significantly boosting experimental efficiency. However, single-cell sequencing is a destructive process, making it impossible to capture the same cell's phenotype before and after perturbation. Consequently, data collected under perturbed and unperturbed conditions are inherently unpaired, creating a critical yet unresolved problem in single-cell perturbation modeling. Moreover, the high dimensionality and sparsity of single-cell expression make direct modeling prone to focusing on zeros and neglecting meaningful patterns. To address these problems, we propose a new paradigm for single-cell perturbation modeling. Specifically, we leverage dual diffusion models to learn the control and perturbed distributions separately, and implicitly align them through a shared Gaussian latent space, without requiring explicit cell pairing. Furthermore, we introduce a sparsity masking strategy in which the mask model learns to predict zero-expressed genes, allowing the diffusion model to focus on capturing meaningful patterns among expressed genes and thereby preserving diversity in high-dimensional sparse data. We introduce **Doloris**, a generative framework that defines a new paradigm for modeling unpaired, high-dimensional, and sparse single-cell perturbation data. It leverages dual conditional diffusion models for separate learning of control and perturbed distributions, complemented by a sparsity masking strategy to enhance prediction of zero-valued genes. The results on publicly available datasets show that our model effectively captures the diversity of single-cell perturbations and achieves state-of-the-art performance. To facilitate reproducibility, we include the code in the supplementary materials. Code available at `https://github.com/ChangxiChi/Doloris`.

## 1 Introduction

Different single-cell perturbations, including CRISPR-based gene knockouts (Barrangou & Doudna, 2016; Lino et al., 2018) and small-molecule treatments (Peidli et al., 2024), act at different layers of cellular mechanisms. Despite significant advancements in sequencing technology, producing perturbation data remains costly and time-consuming. As it is impractical to perform experiments across all cell types and perturbation conditions, accurately predicting perturbation responses under

---

*Corresponding author.

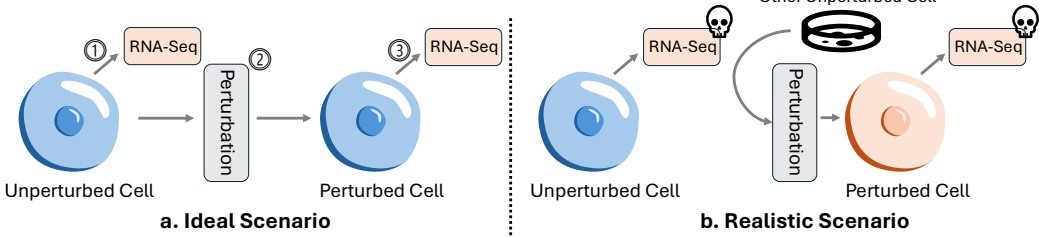

Figure 1: Single-cell perturbation data are unpaired as cells cannot be measured twice.

novel conditions is crucial. This capability significantly enhances biomedical research, particularly in advancing the understanding of gene functions and accelerating drug screening.

RNA-seq requires cell lysis to release RNA during sequencing, making it an irreversible and destructive process for cells (Mortazavi et al., 2008). Consequently, in single-cell perturbation experiments, capturing the same cell's phenotype before and after perturbation is not feasible (Fig. 1). As a result, single-cell perturbation data are fundamentally unpaired. Although existing methods (Roohani et al., 2022; Hetzel et al., 2022b; Bereket & Karaletsos, 2024; Wu et al., 2022; He et al., 2024; Wang et al., 2024; Piran et al., 2024; Chi et al., 2025) for predicting cell responses under unseen perturbation conditions have made significant progress, they often overlook the inherently unpaired nature of single-cell perturbation data, either by forcibly matching samples from the perturbed and unperturbed groups or by disregarding their relationships during modeling. On the other hand, while the unpaired nature of the data has been considered in some studies (Bunne et al., 2023; Cao et al., 2024), their lack of explicit perturbation modeling limits generalization to unseen perturbations.

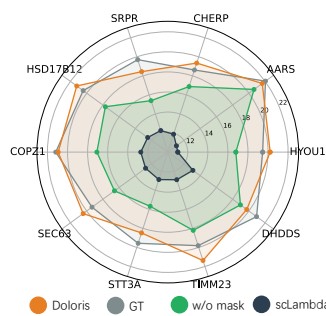

Figure 2: Intra-distance across different model settings. See Section 4.5 for details.

As shown in Fig. 2, directly learning the expression matrix reduces model diversity, as the high dimensionality and sparsity of single-cell data with abundant zero or near-zero values (Xie et al., 2023; Chi et al., 2024) obscures meaningful patterns (Johnstone & Titterington, 2009; Bühlmann & Van De Geer, 2011).

To address these issues, we propose **Doloris** (**D**ual Conditional Diffusi**o**n Imp**l**icit Bridges with Sparsity Masking Strategy f**o**r Unpai**r**ed S**i**ngle-Cell Perturbation E**s**timation), a new paradigm for modeling single-cell perturbations that predicts cellular responses to unseen genetic and molecular perturbations. Inspired by (Su et al., 2022), **Doloris** leverages a dual conditional diffusion (DDIB) framework to model unpaired single-cell perturbation. To address the challenge of unpaired data, it uses a source model for unperturbed cells and a target model for perturbed cells, sharing a latent Gaussian space to implicitly bridge control and perturbed states, while a perturbation specific embedding incorporates gene and molecular perturbation information. Besides, we show that adding more genes reduces the SNR (Fig. 8), indicating that higher dimensionality makes pattern learning harder. On top of this high-dimensional background, single-cell expression is also sparse (Fig. 5). We introduce a sparsity masking strategy that predicts zero-valued genes and steers the diffusion model to focus on expressed signals. Section 4.5 shows that the sparsity masking strategy is effective in mitigating the model's tendency to overfit zeros and preserving diversity.

The main contributions of our work are as follows:

- We introduce **Doloris**, a new paradigm for single-cell perturbation modeling. It explicitly addresses the challenge of unpaired data by learning separate distributions for unperturbed and perturbed cells while maintaining a shared latent space to implicitly bridge control and perturbed distributions, without requiring explicit cell pairing.

- To handle the sparsity and high dimensionality of gene expression, it leverages a sparsity masking strategy that predicts zero-valued genes, ensuring the diffusion model focuses on

meaningful expression patterns instead of abundant zeros. Ablation studies further confirm that the masking strategy effectively mitigates overfitting to zeros and preserving diversity.

- We show that **Doloris** outperforms existing methods across a broad range of evaluation metrics on public genetic and molecular perturbation datasets.

## 2 RELATED WORK AND PRELIMINARIES

### 2.1 PERTURBATION ESTIMATION MODEL

Genetic and molecular perturbations constitute the two main research directions in single-cell perturbation studies. Existing methods have made significant progress in modeling single-cell perturbation responses. Some approaches rely on regression models to predict the outcomes of perturbations (Roohani et al., 2022; Chi et al., 2025; Cheng et al., 2025). Other methods employ generative models to reconstruct the distribution of perturbed states (Lotfollahi et al., 2019; Cui et al., 2024; Hetzel et al., 2022a; Wu et al., 2022; Bereket & Karaletsos, 2024; Wang et al., 2024; Piran et al., 2024). However, many of these approaches largely overlook the intrinsic relationship between control and perturbed samples during modeling. A separate class of methods enforces explicit pairing between unperturbed and perturbed samples, which may introduce unrealistic assumptions about the data.

### 2.2 DIFFUSION PROCESS AND LEARNING OBJECTIVE

In this section, we introduce the basic formulation of diffusion (Luo, 2022; Guo et al., 2023). Given an input sample $x_0$, we progressively add noise to it via the forward diffusion process as follows:

$$x_t = \sqrt{\bar{\alpha}_t} \cdot x_0 + \sqrt{1 - \bar{\alpha}_t} \cdot \epsilon, \epsilon \sim \mathcal{N}(0, \mathbf{I}) \tag{1}$$

where $t \in [0, 1]$ denotes the time step in the diffusion process, and $\bar{\alpha}_t$ is the signal-to-noise ratio at step $t$. The objective of the diffusion model $\epsilon_\theta$ is to predict the true noise from the noisy sample $x_t$. The formula is as follows:

$$\mathcal{L} = \mathbb{E}_{x_0, \epsilon \sim \mathcal{N}(0, \mathbf{I}), t} \left[ \|\epsilon - \epsilon_\theta(x_t, t)\|^2 \right] \tag{2}$$

### 2.3 DDIM INVERSION

The DDIM (Song et al., 2020) proposes a straightforward inversion technique based on the ODE process, which significantly accelerates the inversion of $x_T$ back to $x_0$, based on the assumption that the ODE process can be reversed in the limit of small steps, which can be written as:

$$x_{t-1} = \sqrt{\bar{\alpha}_{t-1}} \left( \frac{x_t - \sqrt{1 - \bar{\alpha}_t} \epsilon_\theta(x_t, t)}{\sqrt{\bar{\alpha}_t}} \right) + \sqrt{1 - \bar{\alpha}_{t-1} - \eta^2} \cdot \epsilon_\theta(x_t, t) + \eta \epsilon_t \tag{3}$$

where $\eta$ determines the stochasticity in the forward process, and $\epsilon_t$ is standard Gaussian noise.

### 2.4 DDIB INFERENCE

Dual Diffusion Implicit Bridges (DDIB,(Su et al., 2022)) provide a mechanism to model transitions between two distributions by learning separate diffusion models $\epsilon_\theta^{(s)}$ and $\epsilon_\theta^{(t)}$ for source and target domains, while connecting them through a shared latent space. Specifically, the process begins by adding noise to sample $x^{(s)}$ from the source distribution as follow:

$$x^{(l)} = \texttt{ODESolve}(x^{(s)}; \epsilon_\theta^{(s)}, 0, 1), \quad \texttt{ODESolve}(x_{t_0}; \epsilon_\theta, t_0, t_1) = x_{t_0} + \int_{t_0}^{t_1} \epsilon_\theta(t, x_t) \, \mathrm{d}t \tag{4}$$

Then, starting from the latent representation $x^{(l)}$, the target diffusion model $\epsilon_\theta^{(t)}$ performs the reverse denoising process to generate a sample $x^{(t)}$ in the target domain:

$$x^{(t)} = \texttt{ODESolve}(x^{(l)}; \epsilon_\theta^{(t)}, 1, 0) \tag{5}$$

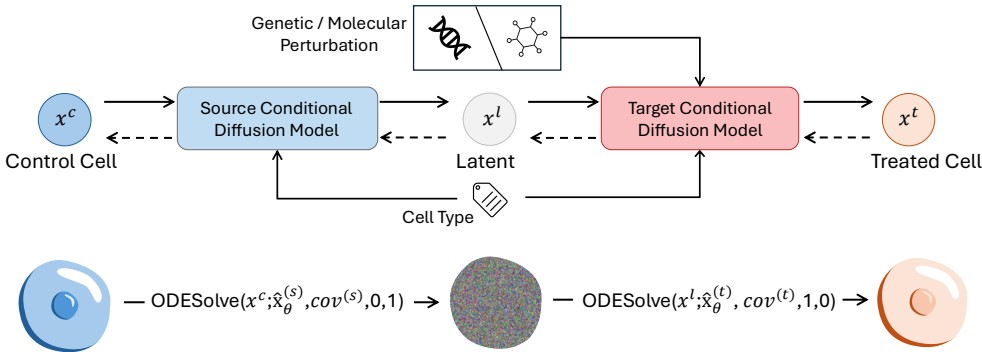

Figure 3: Overview of **Doloris**. **Doloris** predicts cellular responses under unseen perturbation conditions. The source model first maps an unperturbed cell $x^c$ into the shared latent space by applying a DDIM-based forward process conditioned on covariates $cov^{(s)}$, obtaining the latent embedding $x^l$. Conditioned on a given perturbation covariates $cov^{(t)}$, the target model then performs DDIM-based denoising from $x^l$ to generate the predicted perturbed cell $x^t$. For clarity, only the core framework is shown here. The mask model will be introduced later in detail.

## 3 METHODOLOGY

In this section, we introduce our proposed model **Doloris**. The overview is shown in Fig. 3. Specifically, the source model learns the distribution of unperturbed cells, while the target model learns the distribution of cells under various perturbation conditions. By using a source model and a target model that share a prior space, we align the distributions of unperturbed and perturbed cells, thereby addressing the issue of unpaired data. It is worth noting that Fig. 3 illustrates only the core framework. In addition, **Doloris** employs a sparsity masking strategy to predict zero-valued genes. The details of the mask model are presented in Sec. 3.5.

### 3.1 INPUT AND OUTPUT

In the single-cell perturbation prediction task, our goal is to predict the gene expression levels of cells under specific perturbation conditions. During training, the model takes real cell samples as input to learn the transition from the true expression distribution to a Gaussian noise distribution (Section 3.3 and Section 3.4), where the source model is conditioned on $cov^{(s)}$ and the target model is conditioned on $cov^{(t)}$. At the same time, the mask model learns the probabilities of gene activation under perturbation (Section 3.5), which are conditioned solely on $cov^{(t)}$. During inference, control cell sample $x^c$ and condition $cov^{(s)}$ are input to generate a latent embedding $x^l$, which is then denoised under the given perturbation condition $cov^{(t)}$ to output the predicted perturbed gene expression. See Section 3.6 for details.

### 3.2 DATA PREPROCESSING

We first apply the SCANPY package Wolf et al. (2018) to perform log1p normalization on the gene expression data. Here, $N$ represents the dimensionality of the gene expression vector for each single cell. To facilitate stable training, we normalize the gene expression values to the range $[0, 1]$ using the max value $x_{max}$ from the training set after splitting the dataset as: $x' = \frac{x}{x_{\max}}$. When generating predictions, we restore the normalized values back to the original scale by multiplying by $x_{max}$.

### 3.3 SOURCE MODEL FOR THE DISTRIBUTION OF UNPERTURBED CELLS

The source model is a conditional diffusion model designed to capture the gene expression distributions of unperturbed cells. It models the alignment of control cells under different conditions (here represented by cell type labels) with a standard Gaussian latent space.

Unlike conventional diffusion models (Guo et al., 2023), which predict the noise at each time step (Eq. 2), modeling the noise in gene expression data is particularly challenging due to its complexity and weak structure. Therefore, our model directly predicts $x_0$, the clean gene expression data. These two parameterizations are theoretically equivalent because (Luo, 2022):

$$x_0 = \frac{x_t - \sqrt{1 - \bar{\alpha}_t} \epsilon_0}{\sqrt{\bar{\alpha}_t}} \tag{6}$$

Formally, given a control cell sample $x_0^c$ from cell type $ct$ drawn from the unperturbed distribution $p_{ct}^{(s)}$, we obtain the noisy sample $x_t^c$ by applying a forward diffusion process (Eq. 1):

$$x_t^c = \sqrt{\bar{\alpha}_t} \cdot x_0^c + \sqrt{1 - \bar{\alpha}_t} \cdot \epsilon, \epsilon \sim \mathcal{N}(0, \mathbf{I}) \tag{7}$$

The model outputs can be uniformly written as:

$$\hat{x}_0^c = \hat{\mathrm{x}}_\theta^{(s)}(x_t^c, t, cov^{(s)}), \quad cov^{(s)} = \{ct\} \tag{8}$$

Considering the sparsity of gene expression data, we design a mask model, trained independently from the main model, to predict which genes are zero-valued. Consequently, the diffusion model computes the loss only over expressed genes during training. Finally, the diffusion model is trained by minimizing the reconstruction loss between the predicted and true clean gene expression:

$$\mathcal{L}^{(s)} = \mathbb{E}_{x_0^c, t, cov^{(s)}} \left[ \frac{\left\| M^c \odot (x_0^c - \hat{\mathrm{x}}_\theta^{(s)}(x_t^c, t, cov^{(s)})) \right\|^2}{\sum_i M_i^c} \right] \tag{9}$$

here, $M^c$ is a binary mask vector defined as:

$$M_i^c = \begin{cases} 1, & \text{if } x_{0,i}^c \neq 0 \\ 0, & \text{otherwise} \end{cases} \tag{10}$$

## 3.4 TARGET MODEL FOR THE DISTRIBUTION OF PERTURBED CELLS

The target model is largely analogous to the source model, with the main difference being that it learns the distribution of cells under various perturbation conditions, conditioned on both cell type $ct$ and perturbation $P$. Given a treated cell sample $x_0^t$ from cell type $ct$ drawn from the perturbed distribution $p_{ct}^{(t)}$, we obtain the noisy sample $x_t^t$ at timestep $t$ obtained from the perturbed cell $x_0^t$:

$$x_t^t = \sqrt{\bar{\alpha}_t} \cdot x_0^t + \sqrt{1 - \bar{\alpha}_t} \cdot \epsilon, \epsilon \sim \mathcal{N}(0, \mathbf{I}) \tag{11}$$

Considering that perturbations are applied to unperturbed cells to simulate their responses, we need to provide the target model with information about the unperturbed group. However, since the perturbation data is unpaired, we can't directly input a sample from the unperturbed group. Furthermore, using only the expectations $\mu_{ct} \in R^N$ of unperturbed group gene expression $p_{ct}^{(s)}$ is unreasonable, as it disregards cell heterogeneity. During training, random Gaussian noise is added internally to $\mu_{ct}$ with scale $\sigma_{ct}$ (Eq. 12) to preserve cellular heterogeneity. Importantly, this does not assume that gene expression follows a Gaussian distribution, but rather serves as a stochastic mechanism to avoid collapsing to mean profiles.

$$x_{noisy} = \mu_{ct} + \sigma_{ct} \cdot \epsilon, \epsilon \sim \mathcal{N}(0, \mathbf{I}) \tag{12}$$

Finally, the objective of the target model is analogous to the source model, except that it learns from perturbed cells under specific perturbation conditions. Formally, the loss is defined as:

$$\mathcal{L}^{(t)} = \mathbb{E}_{x_0^t, t, cov^{(t)}} \left[ \frac{\left\| M^t \odot (x_0^t - \hat{\mathrm{x}}_\theta^{(t)}(x_t^t, t, cov^{(t)})) \right\|^2}{\sum_i M_i^t} \right] \tag{13}$$

where $M^t$ is a binary mask vector defined in Eq. 10, computed based on the clean sample $x_0^t$. For notational simplicity, we define $cov^{(t)} = \{ct, \mu_{ct}, \sigma_{ct}, P\}$, where $ct$ denotes the cell type, $\mu_{ct}$ and $\sigma_{ct}$ represent the expectation and standard deviation of the corresponding unperturbed distribution $p_{ct}^s$, and $P$ denotes the perturbation.

### 3.5 Sparsity Masking Strategy for Zero-valued Gene Prediction

High-dimensional data are inherently challenging, as learning meaningful patterns becomes increasingly difficult when the number of features grows (Johnstone & Titterington, 2009; Bühlmann & Van De Geer, 2011). In the context of single-cell expression, our analysis shows that as more genes are considered, the relative signal-to-noise ratio (SNR) of the expression data decreases significantly (Fig. 8), indicating that pattern learning becomes more difficult in this high-dimensional space. On top of this high-dimensional challenge, single-cell expression is also sparse (Fig. 5). This sparsity can cause the model to overfit the abundant zeros, obscuring meaningful perturbation-specific patterns. To address this, we introduce a sparsity masking strategy that predicts silenced genes under perturbation, ensuring that the diffusion model focuses on truly expressed genes and learns nontrivial perturbation-specific patterns instead of collapsing to zero-dominant solutions.

Specifically, besides computing the diffusion loss only over expressed genes during model training (Eq. 9 and Eq. 13), the sparsity masking strategy also requires an additional Mask Model $\hat{m}_\theta$ to be trained. This Mask Model predicts which genes are silenced after each perturbation. The output of $\hat{m}_\theta$ can be interpreted as probabilities of gene activation, and the final training objective is to minimize the cross-entropy loss as follow:

$$\mathcal{L}_{mask} = -\frac{1}{N} \sum_{i=1}^{N} \left[ M_i^t \log(\hat{m}_\theta(cov^{(t)})) + (1 - M_i^t) \log(1 - \hat{m}_\theta(cov^{(t)})) \right] \tag{14}$$

here $M^t$ is obtained from the observed cell sample $x_0$ under perturbation $P$ and cell type $ct$ using Eq. 10. However, it only characterizes the predicted marginal distribution. How to derive meaningful samples from this predicted marginal distribution will be discussed in the following section.

### 3.6 Inference

Inference consists of two main steps. First, we generate continuous gene expression values under perturbation conditions using DDIB inference (Sec. 2.4). Second, the Mask Model predicts the expression states of genes after perturbation. In this section, we provide a detailed explanation of how these steps are implemented.

Conditioned on perturbation $P$ and cell type $ct$, we first generate continuous gene expression values using DDIB inference, as illustrated in the lower part of Fig. 3. Specifically, we first randomly sample a control cell $x^c$ from cell type $ct$, and then apply a forward diffusion process to obtain the corresponding latent embedding $x^l$ using source model $\hat{x}_\theta^{(s)}$:

$$x^l = \texttt{ODESolve}(x^c; \hat{x}_\theta^{(s)}, cov^{(s)}, 0, 1) \tag{15}$$

where $cov^{(s)} = \{ct\}$. Then, starting from the latent embedding $x^l$, the target model $\hat{x}_\theta^{(t)}$ performs the denoising process to generate the predicted gene expression profile $x^t$ under perturbation $P$. We note that during inference, we assume a true control cell sample $x^c$ as the starting point. Therefore, unlike during training, there is no need to construct the input using the mean and standard deviation of the unperturbed group. The actual sample $x^c$ is used directly in place of $x_{noisy}$ (Eq. 12).

$$x^t = \texttt{ODESolve}(x^l; \hat{x}_\theta^{(t)}, cov^{(t)}, 1, 0) \tag{16}$$

where $cov^{(t)} = \{ct, x^c, P\}$.

For gene activation prediction, we first feed the control sample $x^c$ with the given condition $cov^{(t)}$ into the Mask Model $\hat{m}_\theta$, which outputs a probability score $p_{\hat{m}_\theta} \in [0,1]^N$ for each gene being active. However, directly drawing independent Bernoulli samples from these probabilities can accumulate severe errors and yield globally inconsistent gene activation patterns. To address this, we propose a more coherent strategy that first identifies training-condition subsets with empirical marginal distributions similar to $p_{\hat{m}_\theta}$, and then updates samples from these subsets according to the predicted probabilities to obtain the final binary activation mask $\hat{M} \in \{0,1\}^N$ (Appendix A.11 for details).

Finally, the prediction is obtained by applying the sparsity mask $\hat{M}$ to the predicted continuous gene expression values $x^t$ via element-wise multiplication, followed by rescaling to the original scale.

$$\hat{x} = (\hat{M} \odot x^t) \times x_{max} \tag{17}$$

### 3.7 IMPLEMENTATION

During training, we separately optimize the dual diffusion models ($\hat{x}_\theta^{(s)}$ and $\hat{x}_\theta^{(t)}$) and the Mask Model $\hat{m}_\theta$. Since the source and target models share the same architecture, with the target model only requiring additional conditioning inputs (e.g., perturbation $P$, etc., see Sec. 3.4), we unify them into a single implementation that jointly handles both $cov^{(s)}$ and $cov^{(t)}$ to simplify training.

The embedding of cell type $ct$ is directly learned as a trainable label representation from the training data, without relying on external models. After receiving perturbation information, the model passes it through a perturbation-specific embedding module, which generates conditional signals for gene and molecular perturbations. Specifically, for gene perturbations, we follow the embedding strategy of (Chi et al., 2025), which enables our model to handle multi-gene knockouts and capture combinatorial perturbation effects. For molecular perturbations, we leverage a pre-trained model (Zhou et al., 2023) to extract molecular representations, which are then used as conditional inputs for the diffusion model to guide generation.

## 4 EXPERIMENTS AND RESULTS

### 4.1 DATASETS

We utilize the Adamson (Adamson et al., 2016) and Norman (Norman et al., 2019) datasets for CRISPR knockouts, and the sci-Plex3 (Srivatsan et al., 2020b) dataset for chemical perturbations. Detailed preprocessing and data splitting procedures are provided in Appendix A.2.

### 4.2 EXPERIMENT SETTINGS

The model is trained using the AdamW (Loshchilov, 2017) optimizer with a learning rate of 0.001 and a batch size of 32. The diffusion process is configured with a total of 500 steps. For inference, we adopt DDIM (Song et al., 2020) sampling with 50 steps. For datasets Adamson, Norman and SciPlex3, training steps are adjusted to $10,000$, $10,000$ and $100,000$, respectively. All our method and its competitors are conducted using one Nvidia A100 80G GPU.

### 4.3 DOLORIS OUTPERFORM EXISTING METHODS

For evaluation, we observe strong heterogeneity in single-cell data, where many differentially expressed (DE) genes exhibit bimodal distributions under the same condition (Fig. 4). This limitation renders expectation-based metrics unreliable. In particular, for bimodal gene expression distributions, the conditional mean is not biologically meaningful, and metrics such as RMSE computed on the mean fail to capture the true distributional characteristics. To address this, we introduce Energy Distance (E-distance) and Earth Mover's Distance (EMD). E-distance captures overall distributional alignment by considering both inter-group and intra-group distances, while EMD quantifies gene-level shifts by measuring the minimal cost to align predicted and true distributions. Together, they provide a comprehensive and robust assessment of model perfor-

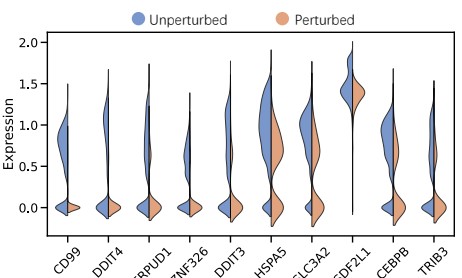

Figure 4: Under the same experimental condition, many genes show a bimodal distribution. The figure shows top DE genes for ZNF326 knockout versus unperturbed cells.

mance at both the population and gene levels. Detailed computation procedures are provided in the Appendix A.6.

Table 1 shows that **Doloris** outperforms GEARS (Roohani et al., 2022), graphVCI (Wu et al., 2022), scGPT (Cui et al., 2024), BioLord (Piran et al., 2024), and GRAPE (Chi et al., 2025) across most evaluation metrics, all of which rely on forced pairing of perturbed and unperturbed cells during training. For the regression models, this setup tends to bias learning toward the mean of the data, preventing the capture of the heterogeneity of single-cell gene expression profiles. Methods such as

Table 1: Comparisons on Adamson and sci-Plex3 datasets. Metrics include RMSE, E-distance, and EMD computed on all and top-20/40 DE genes. Tasks correspond to unseen single-gene and drug–cell line-dosage perturbations, respectively.

| Model | All | | | DE20 | | | DE40 | | |
|---|---|---|---|---|---|---|---|---|---|
| | RMSE(↓) | E-distance(↓) | EMD(↓) | RMSE(↓) | E-distance(↓) | EMD(↓) | RMSE(↓) | E-distance(↓) | EMD(↓) |
| **- Unseen single genetic perturbation prediction results** | | | | | | | | | |
| **Doloris** | **0.0336** | **0.4682** | **0.0348** | **0.1094** | **0.4653** | **0.0789** | **0.0987** | **0.4976** | **0.0811** |
| | ±0.0103 | ±0.1398 | ±0.0033 | ±0.0506 | ±0.1930 | ±0.0514 | ±0.0578 | ±0.1847 | ±0.0452 |
| ScLambda | 0.0505 | 1.9939 | 0.0906 | 0.2539 | 0.6996 | 0.0914 | 0.2197 | 0.7229 | 0.0950 |
| | ±0.0257 | ±0.1296 | ±0.0040 | ±0.0192 | ±0.2997 | ±0.0476 | ±0.0589 | ±0.2615 | ±0.0389 |
| GRAPE | 0.0510 | 0.8705 | 0.0444 | 0.1850 | 0.7514 | 0.1528 | 0.1697 | 0.7648 | 0.1503 |
| | ±0.0110 | ±0.0484 | ±0.0024 | ±0.0066 | ±0.0523 | ±0.0234 | ±0.0047 | ±0.0565 | ±0.0182 |
| GEARS | 0.0544 | 0.8921 | 0.0531 | 0.1759 | 0.7884 | 0.1298 | 0.1781 | 0.7935 | 0.1221 |
| | ±0.0088 | ±0.1304 | ±0.0027 | ±0.0078 | ±0.1245 | ±0.0324 | ±0.0054 | ±0.1273 | ±0.0231 |
| scGPT | 0.5372 | 2.6318 | 0.1724 | 0.7151 | 1.2571 | 0.3895 | 0.7021 | 1.4484 | 0.3781 |
| | ±0.1482 | ±0.0441 | ±0.0355 | ±0.1246 | ±0.3373 | ±0.1032 | ±0.2207 | ±0.3087 | ±0.0866 |
| linear | 0.0473 | 0.8658 | 0.0373 | 0.2143 | 0.8583 | 0.1702 | 0.2007 | 0.8958 | 0.1631 |
| | ±0.0008 | ±0.0251 | ±0.0024 | ±0.0068 | ±0.0525 | ±0.02265 | ±0.0040 | ±0.0429 | ±0.0199 |
| **- Unseen molecular perturbation prediction results** | | | | | | | | | |
| **Doloris** | **0.0287** | **0.4055** | **0.0265** | **0.0625** | **0.2484** | **0.0743** | **0.0547** | **0.2406** | **0.0649** |
| | ±0.0157 | ±0.2190 | ±0.0051 | ±0.0412 | ±0.1710 | ±0.0216 | ±0.0460 | ±0.1671 | ±0.0219 |
| BioLord | 0.0409 | 1.2739 | 0.0703 | 0.1094 | 0.8823 | 0.2157 | 0.0945 | 1.0314 | 0.1920 |
| | ±0.0180 | ±0.1947 | ±0.0103 | ±0.0622 | ±0.1529 | ±0.0645 | ±0.0497 | ±0.1451 | ±0.0477 |
| chemCPA | 0.0570 | 0.7847 | 0.0838 | 0.1462 | 0.4717 | 0.1836 | 0.1314 | 0.5008 | 0.1784 |
| | ±0.0130 | ±0.1029 | ±0.0081 | ±0.0271 | ±0.1571 | ±0.0358 | ±0.0167 | ±0.1659 | ±0.0261 |
| CPA | 0.0697 | 0.9894 | 0.1357 | 0.2006 | 0.9737 | 0.3761 | 0.1807 | 1.0794 | 0.3856 |
| | ±0.0253 | ±0.1336 | ±0.0461 | ±0.0935 | ±0.9768 | ±0.0667 | ±0.0667 | ±1.1890 | ±0.0387 |
| GraphVCI | 0.6212 | 0.8393 | 0.0986 | 0.5886 | 0.4958 | 0.2016 | 0.6007 | 0.5174 | 0.1861 |
| | ±0.0772 | ±0.1823 | ±0.0108 | ±0.1441 | ±0.1275 | ±0.0379 | ±0.1231 | ±0.1347 | ±0.0288 |

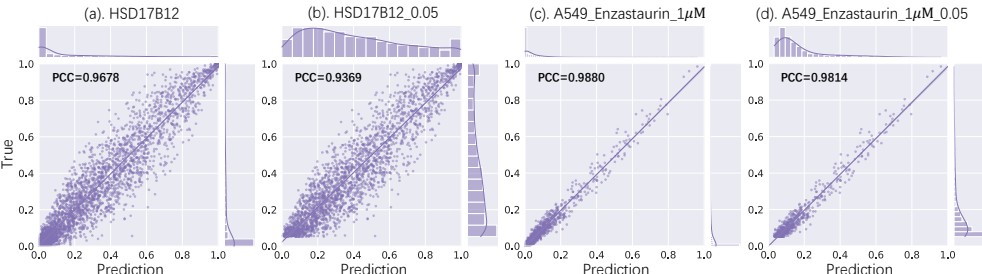

Figure 5: Visualization of predicted gene activation probabilities by the Mask Model. Panels (a) and (c) show the results under perturbation of HSD17B12 and A549-Enzastaurin-1$\mu$M (cell type-drug-dosage), respectively, while panels (b) and (d) display only genes with predicted activation probabilities greater than 0.05 from (a) and (c).

CPA (Lotfollahi et al.), chemCPA (Hetzel et al., 2022a) and scLambda (Wang et al., 2024) further underperform because some of them reconstruct only perturbed cells without explicitly modeling the transition from the unperturbed state. Notably, while linear baseline (Ahlmann-Eltze et al., 2025) achieves competitive performance against some deep learning models, it fundamentally lacks the capacity to model complex, non-linear distributional shifts. More importantly, the modeling paradigm employed by **Doloris** overcomes the challenges posed by high-dimensional and sparse single-cell data. By leveraging an additional sparsity masking strategy, the diffusion model can focus on expressed genes rather than fitting zero-valued entries, thereby capturing more biologically relevant information (Fig. 2). Crucially, **Doloris** addresses the limitations of paired data by employing dual implicit bridges, which explicitly and flexibly model the relationship between unperturbed and perturbed states.

To validate the effectiveness of the Mask Model, Fig. 5 shows that it achieves good performance in predicting gene activation probabilities under different perturbation conditions. We compared the predicted gene activation probabilities for all genes with the empirical probabilities for genes with expression levels greater than 0.05.

### 4.4 DOLORIS PERFORMS WELL ON OOD DRUG AND DOUBLE GENE PERTURBATION

To further validate the effectiveness of **Doloris**, we evaluate its performance on double gene knock-outs using the Norman dataset (Norman et al., 2019) and on out-of-distribution (OOD) drugs (described in Sec. 4.1) in Tab. 2. Double gene knockouts involve complex gene–gene interactions, and experimental results show that our model effectively captures these interactions. To predict the effects of double gene perturbations, we use all observed samples under single gene perturbations and unperturbed conditions as the training set. It has been previously demonstrated that OOD drugs (Srivatsan et al., 2020a; Hetzel et al., 2022a), which were not observed during training, predominantly target epigenetic regulation, tyrosine kinase signaling, and cell cycle regulation. These drugs are representative of key biological processes and are often distinct from the drug in the training set. Our model demonstrates superior performance, suggesting that it better captures the effects of unseen molecules on cellular behavior. Importantly, our design allows us to infer the effects of previously unseen drug molecules as well as unobserved gene perturbations.

Table 2: Evaluation of model performance on double gene (Norman) and OOD drug perturbations (sci-Plex3). We highlights the top two methods in red and orange, respectively.

| - Double gene perturbations | | | | | |
|---|---|---|---|---|---|
| | **Doloris** | linear | GRAPE | GEARS | ΔScore |
| RMSE All | **0.0385** ±0.0129 | 0.0405 ±0.0001 | 0.0516 ±0.0187 | 0.0533 ±0.0079 | +0.0020 |
| RMSE DE20 | **0.2431** ±0.0828 | 0.2523 ±0.0108 | 0.2871 ±0.0629 | 0.3095 ±0.04222 | +0.0092 |
| RMSE DE40 | **0.2095** ±0.0678 | 0.2123 ±0.0174 | 0.2947 ±0.0846 | 0.3284 ±0.0525 | +0.0028 |
| E-distance All | **0.6819** ±0.1232 | 0.7886 ±0.0611 | 0.7862 ±0.0899 | 1.1204 ±0.0206 | +0.1043 |
| E-distance DE20 | **0.7888** ±0.1277 | 0.8276 ±0.0637 | 0.9272 ±0.0806 | 0.8665 ±0.0213 | +0.0388 |
| E-distance DE40 | **0.8143** ±0.1524 | 0.8835 ±0.0651 | 0.9601 ±0.0842 | 0.9614 ±0.0163 | +0.0692 |
| EMD All | **0.0179** ±0.0039 | 0.0190 ±0.0020 | 0.0289 ±0.0019 | 0.0306 ±0.0033 | +0.0011 |
| EMD DE20 | **0.2025** ±0.0825 | 0.2175 ±0.0314 | 0.2385 ±0.0381 | 0.2403 ±0.0304 | +0.0150 |
| EMD DE40 | **0.1678** ±0.0694 | 0.1857 ±0.0256 | 0.1978 ±0.0304 | 0.2347 ±0.0246 | +0.0179 |
| - OOD molecular perturbations | | | | | |
| | **Doloris** | chemCPA | GraphVCI | - | |
| RMSE All | **0.0547** ±0.0305 | 0.0689 ±0.0150 | 0.5431 ±0.0852 | - | +0.0142 |
| RMSE DE20 | **0.1549** ±0.1131 | 0.2902 ±0.0690 | 0.3387 ±0.1685 | - | +0.1353 |
| RMSE DE40 | **0.1313** ±0.0910 | 0.2489 ±0.0263 | 0.3968 ±0.1331 | - | +0.1176 |
| E-distance All | **0.7071** ±0.1298 | 0.8861 ±0.0678 | 0.8468 ±0.1914 | - | +0.1397 |
| E-distance DE20 | **0.4744** ±0.1876 | 0.7377 ±0.2248 | 0.7123 ±0.1945 | - | +0.2379 |
| E-distance DE40 | **0.4839** ±0.1643 | 0.7710 ±0.2004 | 0.8469 ±0.1914 | - | +0.2871 |
| EMD All | **0.0295** ±0.0088 | 0.0959 ±0.0096 | 0.0986 ±0.0121 | - | +0.0664 |
| EMD DE20 | **0.1305** ±0.0611 | 0.3435 ±0.0761 | 0.3163 ±0.0631 | - | +0.1858 |
| EMD DE40 | **0.1071** ±0.0514 | 0.3004 ±0.0745 | 0.2776 ±0.0500 | - | +0.1705 |

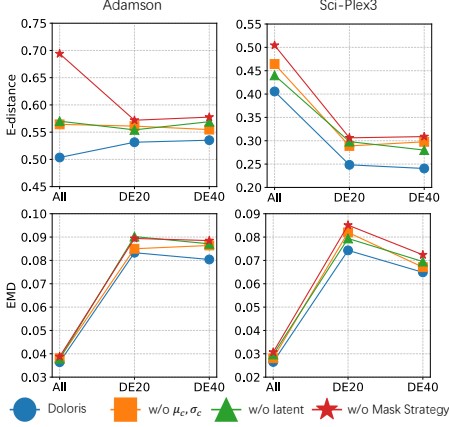

Figure 6: Ablation study results.

### 4.5 ABLATION STUDY

To further evaluate the effectiveness of **Doloris**, we compare it with the following methods through an ablation study. 1)**w/o** $\mu_c, \sigma_c$: Excludes the mean and variance of the unperturbed group from the model input. 2)**w/o latent**: During sampling, the input latent embedding $x^l$ in Eq. 16. b is replaced with random Gaussian noise. 3)**w/o mask model**: Removing the mask model forces the model to predict the expression of all genes during training. The results are shown in Fig.6.

The experimental results indicate that the $\mu_c, \sigma_c$ of unperturbed cells are crucial, as perturbations essentially represent a transition from the unperturbed state. Compared to random Gaussian noise, latent embeddings generated by adding noise to unperturbed cells provide a more structured and interpretable initialization, leading to significantly improved generation quality and modeling efficiency. Experimental results highlight the critical role of the mask model. Due to the sparsity of gene expression data, with many zero-valued genes, models without masking tend to focus on predicting zeros, which diverts attention from actively expressed genes and reduces both diversity and biological fidelity in the generated profiles. As shown in Fig. 2, the intra-class distances (Eq. 20) of predictions decrease in models trained without masking strategy.

## 5 CONCLUSION

In this work, we present **Doloris**, a novel paradigm for single-cell perturbation modeling that explicitly addresses the challenges of unpaired data. By leveraging a dual conditional diffusion framework, our approach aligns the distributions of unperturbed and perturbed cells without requiring explicit sample pairing, while a perturbation-specific embedding module provides genetic and molecular level conditional signals. To handle the sparsity and high dimensionality of single-cell gene expression, we introduce a mask model that predicts zero-valued genes, ensuring that the model focuses

on biologically meaningful signals and preserves diversity. Furthermore, we propose a biologically grounded evaluation metric that captures cellular heterogeneity and the diversity of single-cell responses. Experimental results on genetic and molecular perturbation datasets demonstrate that **Doloris** outperforms existing methods and generalizes to unseen perturbations. Our work establishes a new modeling paradigm for single-cell perturbation, enabling more accurate and biologically faithful predictions of cellular responses under novel conditions.

## 6 ACKNOWLEDGMENTS

This work was supported by National Science and Technology Major Project (No. 2022ZD0115101), National Natural Science Foundation of China Project (No. 623B2086), National Natural Science Foundation of China Project (No. U21A20427), Project (No. WU2022A009) from the Center of Synthetic Biology and Integrated Bioengineering of Westlake University, and the Zhejiang Province Selected Funding for Postdoctoral Research Projects (ZJ2025113).

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

# A APPENDIX

## A.1 USE OF LLM

The LLM only assisted us in checking spelling and grammar.

## A.2 DATASETS DETAILS

**Adamson** This dataset contains 87 types of single-gene perturbations in a single cell type. We perform single-gene perturbation prediction on this dataset. For data splitting, 30% of the perturbation conditions are randomly selected as the test set, while the remaining perturbations and control cells are used for training. Data preprocessing follows the procedures described in (Chi et al., 2025).

**Norman** This dataset includes both single-gene and double-gene perturbations. In our study, we focus on predicting double-gene perturbations. For data splitting, all control cells and single-gene perturbations are used as the training set, while all double-gene perturbations are reserved for the test set. Data preprocessing follows the procedures described in (Chi et al., 2025).

**sci-Plex3** We use it to evaluate model performance on out-of-distribution (OOD) drugs and on unseen combinations of cell type, drug, and dosage. The dataset comprises experiments on three cell lines treated with 188 drugs, each at four dosages. For data splitting, we first designate all samples under certain drug conditions as the OOD (Out-of-Distribution) test set, based on prior analyses reported in Srivatsan et al. (2020a); Hetzel et al. (2022a). For the remaining data, all control group cells are included in the training set. Then, for each experimental condition defined by a unique combination of drug, dosage, and cell type, the corresponding group of cells is assigned to the test set with a 30% probability, and to the training set otherwise. Data preprocessing follows the procedures described in (Hetzel et al., 2022a).

## A.3 EFFECT OF THE NUMBER OF FUNCTION EVALUATIONS (NFE)

In addition, we evaluated the effect of the Number of Function Evaluations (NFE) of diffusion sampling steps on reconstruction performance. As shown in Fig. 7, using a single step results in drastically worse reconstruction metrics, highlighting that multi-step denoising is essential. Increasing the number of steps to 30 yields substantial improvements, while further increasing to 50 or 70 steps provides only marginal gains. Thus we adopt 50 steps to balance reconstruction quality and computational efficiency.

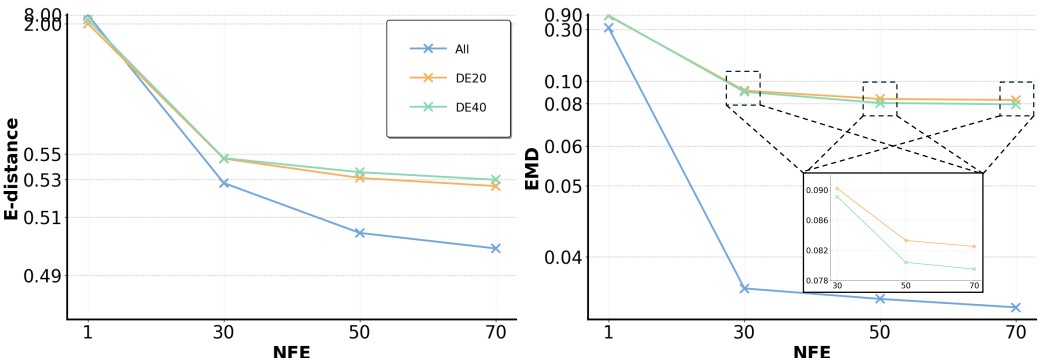

Figure 7: Performance across different NFE.

## A.4 COMPUTATIONAL COST

Assuming that the forward and reverse diffusion processes each incur a computational cost of $N$ (corresponding to the number of DDIM sampling steps) and that the Mask Model introduces a lightweight cost of 1, the theoretical cost per generated sample is $2N + 1$.

In the context of large-scale perturbation generation, where there is a single control cell type and $k$ distinct perturbation conditions with $n$ samples per condition, we can significantly reduce computation by reusing the latent representations obtained from the forward diffusion process rather than regenerating them for each sample. Letting $m$ denote the number of control cells used in the forward process, the total computational cost is then given by $(1 + N) \cdot k \cdot n + m \cdot N$.

For moderate $m$ and large $k$, this cost is comparable to that of a standard diffusion model which directly denoises $k \cdot n$ samples, $N \cdot k \cdot n$. This indicates that when generating a large number of samples, our method has a computational cost similar to standard diffusion.

## A.5 CURSE OF DIMENSION

In this experiment, we select the top 50, 100, 200, 500, 1000 and 2000 highly variable genes (HVGs) and compute the signal-to-noise ratio (SNR) for each perturbation type using only these genes. To better highlight the relative differences across gene sets, the SNR values are normalized relative to the top 50 genes.

SNR is defined as the ratio of the between-condition variance to the within-condition variance:

$$\text{SNR} = \frac{\frac{1}{C} \sum_{c=1}^{C} \|\mu_c - \mu_{\text{overall}}\|_2^2}{\frac{1}{N} \sum_{i=1}^{N} \|x_i - \mu_{y_i}\|_2^2 + \epsilon}, \tag{18}$$

where $C$ is the number of perturbation conditions, $N$ is the total number of cells, $\mu_c$ is the mean expression vector for condition $c$, $\mu_{\text{overall}}$ is the overall mean across all cells, $x_i$ is the expression vector of cell $i$, $y_i$ is its condition label, and $\epsilon$ is a small constant to prevent division by zero.

The Relative SNR defined as:

$$\text{Relative SNR}_{\text{top k}} = \frac{\text{SNR}_{\text{top k}}}{\text{SNR}_{\text{top 50}}} \tag{19}$$

Moreover, after masking out values equal to 0, the resulting data exhibit a substantially improved signal-to-noise ratio compared to the original measurements. If a generative model is trained directly on the raw data, the excessively low signal-to-noise ratio may prevent it from effectively distinguishing random noise from genuine underlying signals.

## A.6 EVALUATION METRIC

In this section, we introduce two metrics—Energy Distance (E-distance) and Earth Mover's Distance (EMD)—which we propose to better quantify the prediction performance of single-cell perturbation models. Given the prediction $X = X_1, X_2, \ldots, X_n \in \mathbb{R}^{n \times N}$ and the true samples $Y = Y_1, Y_2, \ldots, Y_m \in \mathbb{R}^{m \times N}$, where $n$ and $m$ denote the number of cells and $D$ the number of genes.

The E-Distance between $X$ and $Y$ is defined as:

$$D_E(X,Y) = \underbrace{\frac{2}{nm} \sum_{i=1}^{n} \sum_{j=1}^{m} \|X_i - Y_j\|_2}_{2 \times \text{inter-class distance}} - \underbrace{\frac{1}{n^2} \sum_{i=1}^{n} \sum_{j=1}^{n} \|X_i - X_j\|_2}_{\text{intra-class distance (X)}} - \underbrace{\frac{1}{m^2} \sum_{i=1}^{m} \sum_{j=1}^{m} \|Y_i - Y_j\|_2}_{\text{intra-class distance (Y)}}$$
$$\tag{20}$$

where $\|\cdot\|_2$ denotes the Euclidean norm.

Different from the traditional formulation of Earth Mover's Distance (EMD) based on optimal transport, we adopt a practical implementation that averages the one-dimensional Wasserstein distances across gene dimensions. Specifically, the EMD between $X$ and $Y$ is calculated as:

$$D_{EMD}(X,Y) = \frac{1}{|N|} \sum_{j \in N} \text{EMD}(X_{:,j}, Y_{:,j}), \tag{21}$$

where $X_{:,j} \in \mathbb{R}^n$ and $Y_{:,j} \in \mathbb{R}^m$ denote the predicted and true expression values of gene $j$ across all cells, respectively. Each $\text{EMD}(X_{:,g}, Y_{:,g})$ is computed as the 1D Wasserstein distance between the marginal distributions of gene $g$.

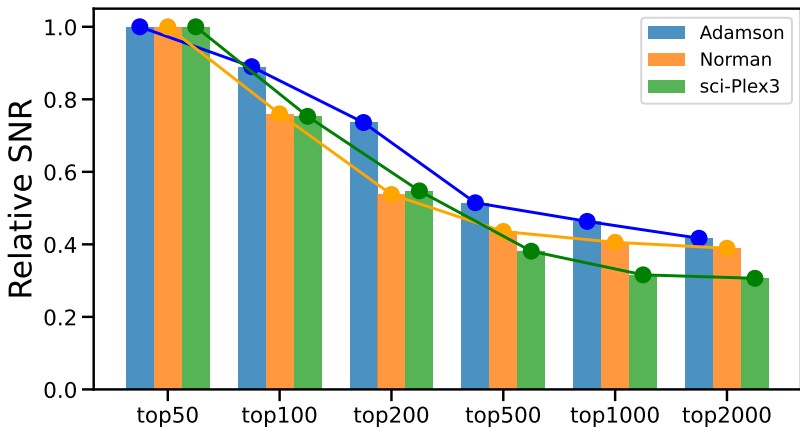

Figure 8: Trend of Relative SNR as more genes are included. Details of the calculation are provided in the Appendix A.5.

In summary, our evaluation framework integrates E-distance for population-level structure and EMD for individual gene-level accuracy, ensuring a robust and comprehensive assessment.

### A.7 PERTURBATION MODELING

To model cellular perturbations, we leverage prior biological knowledge in the form of a gene regulatory network (GRN) $G \in \{0,1\}^{N \times N}$, which is represented as an unweighted graph capturing relationships among genes. Within our model, a Graph Attention Network (GAT) is applied to the GRN to generate a gene embedding, which acts as the perturbation representationfor downstream prediction.

$$f = \text{GAT}(G) \qquad (22)$$

We chose GAT because its attention mechanism allows adaptive weighting of gene–gene interactions, which is particularly useful for modeling regulatory effects under perturbations. To construct the initial node features, we aggregated the gene expression data from the training set. Considering the computational intractability of using individual cell samples directly, we computed the expectation of gene expression for all samples within each perturbation condition. The initial node feature matrix ($\in R^{N \times K}$) was then formed by concatenating these condition-specific expectations ($\in R^{N \times N_P}$), followed by Principal Component Analysis (PCA) for dimensionality reduction (Ahlmann-Eltze et al., 2025). These initialized features serve as static priors and remain frozen (non-trainable) throughout the entire training process.

Unlike random initialization, which treats genes as indistinguishable entities lacking semantic context, our data-driven initialization explicitly incorporates the intrinsic biological properties and expression patterns of each gene. This ensures that the model starts with a biologically meaningful representation space, rather than learning from scratch.

For gene perturbations, we first perform GAT message passing to aggregate regulatory information across the graph. Subsequently, we extract the updated node embedding of the specific perturbed gene from the aggregated graph representations. This context-aware embedding is then utilized as the perturbation condition. For molecular perturbations, we first extract molecular embeddings using a pretrained molecule model (Zhou et al., 2023). These embeddings are then combined with associated treatment information, such as dosage, to form a condition-specific perturbation vector.

### A.8 RELATION WITH HURDLE MODELS

Our sparsity masking strategy can be interpreted as a Hurdle Model applied at the single-cell level: the first component models the probability of gene activation, and the second models the expression values for active genes, while explicitly preserving global dependencies across genes.

### A.9 Zero Expression Carry Biological Meaning in scRNA-seq Data

Numerous studies have shown that dropout is not purely random, and many observed zeros carry meaningful biological informationChoi et al. (2020); Qiu (2020); Jiang et al. (2022). Therefore, leveraging gene–gene dependencies to predict gene activation states is fully justified in this setting and does not constitute a "big claim".

### A.10 Other Related Work

Although (Bunne et al., 2023; Dong et al., 2023) also address unpaired data, their task assumes access to both pre- and post-perturbation cells and focuses on finding optimal pairings between them. In contrast, our task is to predict the post-perturbation state directly from the control cells and the perturbation condition, which is fundamentally different.

### A.11 Mask Model Prediction Strategy

Let the model predict gene activation probabilities from $\hat{\mathrm{m}}_\theta$ for a cell as $p_{\hat{\mathrm{m}}_\theta} = (p_1, p_2, \ldots, p_N)$. A naive independent Bernoulli sampling would give:

$$\hat{M}_i \sim \text{Bernoulli}(p_i), \quad i = 1, \ldots, N, \tag{23}$$

which often leads to globally inconsistent masks. To address this, we purpose a solution as follow.

For a given perturbation condition, we first identify a reference subset of training cells

$$S_{\text{cond}} \in \mathcal{D}_{\text{cond}} \tag{24}$$

here, $\mathcal{D}$ denotes the entire training dataset containing all cells under various perturbation conditions. $\mathcal{D}_{\text{cond}} \subset \mathcal{D}$ represents all observed cells under a specific perturbation condition. $S_{\text{cond}}$ denotes a sample from $\mathcal{D}_{\text{cond}}$.

The empirical marginal activation distributions of cells in the reference subset $\{q_i^{S_{\text{cond}}}\}_{i=1}^N$ are required to be as close as possible to the model-predicted probabilities $p_{\hat{\mathrm{m}}_\theta}$. Formally, the reference subset $S_{\text{cond}}$ is selected from all cells under the same perturbation condition $\mathcal{D}_{\text{cond}}$ by minimizing the Euclidean (L2) distance between the predicted probabilities and the empirical marginal distributions:

$$S_{\text{cond}}^* = \arg \min_{S' \subset \mathcal{D}_{\text{cond}}} \left\| p_{\hat{\mathrm{m}}_\theta} - \{q_i^{S'}\}_{i=1}^N \right\|_2. \tag{25}$$

From the selected reference subset $S_{\text{cond}}^*$, we randomly sample a real sample:

$$\tilde{s} \sim \text{UniformSample}\Big(\{s \mid s \in S_{\text{cond}}\}\Big) \tag{26}$$

and then we obtain real mask sample $\tilde{m}$ by applying Eq. 10 on $\tilde{s}$.

We then update the sampled mask $\tilde{m}$ according to the predicted probabilities $p_{\hat{\mathrm{m}}_\theta}$ using high and low thresholds $\delta_h$ and $\delta_l$, which are set to 0.95 and 0.05, respectively:

$$\hat{M}_i = \begin{cases} 1, & \text{if } p_{\hat{\mathrm{m}}_\theta, i} \geq \delta_h, \\ 0, & \text{if } p_{\hat{\mathrm{m}}_\theta, i} \leq \delta_l, \\ \tilde{m}_i, & \text{otherwise,} \end{cases} \tag{27}$$

Finally, the coherent binary mask for the cell is

$$\hat{M} = (\hat{M}_1, \hat{M}_2, \ldots, \hat{M}_N) \in \{0, 1\}^N. \tag{28}$$

