# OpenReview forum: "Doloris: Dual Conditional Diffusion Implicit Bridges with Sparsity Masking Strategy for Unpaired Single-Cell Perturbation Estimation"
_ICLR.cc/2026/Conference — ICLR 2026 Poster_

### Official Review · Reviewer_DS5U · 2025-10-25

**Soundness:** 2
**Presentation:** 2
**Contribution:** 2
**Rating:** 2
**Confidence:** 3

**Summary:**

This paper presents Doloris, a generative framework designed to predict single-cell perturbation responses from unpaired data. It uses a dual conditional diffusion implicit bridge (DDIB) to learn separate distributions for control and perturbed cells while connecting them via a shared latent space. To address sparsity, it introduces a separate masking model to predict which genes will be silent, allowing the main diffusion model to focus on modeling active gene signals. The authors demonstrate performance on several public datasets using distributional metrics.

**Strengths:**

The DDIB-based approach is a reasonable way to model the transition between control and perturbed states without enforcing explicit pairings.

**Weaknesses:**

- The motivation is unclear, and I don’t think the current motivation has a fair acknowledgement of existing methods. Multiple OT-based perturbation models [1-3], and STATE considers the same problem [4], as well as CellFlow [5]. Variational inference models directly consider sparsity and count distribution [6].
- I don’t think there are anything related to a negative binomial in your method. The method has nothing to do with ZINB at the moment; therefore, the discussion in Appendix A.6 appears problematic.
- It is unclear how the results compare to well-established benchmarks in [4, 7]. In particular, linear baselines or simple averaging have shown better performance compared to deep-learning methods in these cases.
- The paper lacks an ablation study on the value of multi-step inference versus a one-step approach.
- More comprehensive metrics and evaluations should be employed, e.g., the Cell-eval metrics [4].
- It is unclear how the ground truth in Fig. 5 is derived, and how the authors distinguish biological and technical zeros.

[1] Bunne, Charlotte, et al. "Learning single-cell perturbation responses using neural optimal transport." Nature methods 20.11 (2023): 1759-1768.
[2] Klein, Dominik, et al. "Mapping cells through time and space with moscot." Nature 638.8052 (2025): 1065-1075.
[3] Dong, Mingze, et al. "Causal identification of single-cell experimental perturbation effects with CINEMA-OT." Nature methods 20.11 (2023): 1769-1779.
[4] Adduri, Abhinav K., et al. "Predicting cellular responses to perturbation across diverse contexts with State." bioRxiv(2025): 2025-06.
[5] Klein, Dominik, et al. "CellFlow enables generative single-cell phenotype modeling with flow matching." bioRxiv (2025): 2025-04.
[6] Weinberger, Ethan, Chris Lin, and Su-In Lee. "Isolating salient variations of interest in single-cell data with contrastiveVI." Nature Methods 20.9 (2023): 1336-1345.
[7] Ahlmann-Eltze, Constantin, Wolfgang Huber, and Simon Anders. "Deep-learning-based gene perturbation effect prediction does not yet outperform simple linear baselines." Nature Methods (2025): 1-5.

**Questions:**

See weaknesses.

---

> ### Author Response · Authors · 2025-11-16
> **Response to Reviewer DS5U (1/2)**
>
> Thanks for your careful reviews!
>
> >  Question about motivation and existing work
>
> **Answer:** Thank you for the suggestion.
>
> # **existing work:**
>
> We already mention some related work in Section 1, but we will expand this part to clearly compare our method with these approaches:
>
> Although [1–3] also address unpaired data, **their task assumes access to both pre- and post-perturbation cells** and focuses on **finding optimal pairings between them.** In contrast, **our task is to predict the post-perturbation state directly from the control cells and the perturbation condition**, which is fundamentally different.
>
> As for the **concurrent works,** STATE[4] relies on Set Transformer to learn a set-to-set mapping. This approach is indeed milder than enforcing one-to-one cell pairing, but **it still violates the unpaired assumption: the true correspondence between sets is unknown**, and the model **must rely on artificially constructed set pairings.** CellFlow[5] simply applies flow matching. However, **flow matching itself is not designed for unpaired transitions**: during training, it still **requires random pairing between nodes from the source and target distributions[7].** Therefore, it also violates the unpaired assumption. However, Doloris **avoids any form of pairing** by training separate diffusion models for control and perturbed conditions and **bridging them through a shared Gaussian latent space.**
>
> Moreover, we acknowledge that [6] consider sparsity and count distribution. **However, when determining whether a gene is active, it assumes independent Bernoulli variables for each gene.** In high-dimensional gene expression data, this **ignores global gene-gene dependencies, leading to error accumulation across genes.** In contrast, our mask model explicitly accounts for global structure (Sec A.7).
>
> # **motivation:**
> We would like to clarify that our framework’s ability to address core challenges in single-cell perturbation modeling as follow:
>
> 1. To handle unpaired perturbation data, our framework trains separate diffusion models for control and perturbed conditions, **bridging them via a shared Gaussian latent space without any pairing.** This enables effective distribution-level learning from unpaired data, addressing a fundamental challenge in single-cell perturbation modeling.
>
> 2. We predict gene activation separately using a mask model, which **preserves the global structure of gene expression activation state during sampling** and **avoids the error accumulation caused by independent Bernoulli assumptions** in high-dimensional, sparse single-cell data.
>
> >  Question about negative binomial
>
> **Answer:** Our method indeed draws inspiration from the ZINB in the sense that we separately model discrete gene activation and continuous expression levels. However, unlike ZINB-based methods, we do not explicitly assume a negative binomial or zero-inflated negative binomial distribution. We will revise Appendix A.6 to clarify this point and avoid any potential misunderstanding.
>
> >  One-step generation？
>
> **Answer:** It is well-known that diffusion models produce significantly lower-quality outputs when forced into one-step generation[8]. Consistently, in our experiments, forcing Doloris to generate in a single step leads to dramatically worse reconstruction metrics, confirming that multi-step denoising is essential for high-dimensional single-cell data.
>
> | Method     | E_distance_all | EMD_all | E_distance_DE20 | EMD_DE20 | E_distance_DE40 | EMD_DE40 |
> |------------|----------|----------|----------|----------|----------|----------|
> | 1 step      |7.6654±0.1036|0.7623 ±0.1954|2.0576 ±0.2685|0.7623 ±0.1954|2.8376±0.3134|0.8192 ±0.1801|
> | 30 step      |0.5279±0.1532|0.0372 ±0.0036|0.5459 ±0.2390|0.0852 ±0.0372|0.5463±0.2071|0.0831 ±0.0395|
> | 50 step    |0.5035 ±0.1698|0.0364 ±0.0038|0.5312 ±0.1937|0.0833 ±0.569|0.5352 ±0.2085|0.0804 ±0.0481|
>
> [1] Bunne, Charlotte, et al. "Learning single-cell perturbation responses using neural optimal transport." Nature methods 20.11 (2023): 1759-1768.
>
>  [2] Klein, Dominik, et al. "Mapping cells through time and space with moscot." Nature 638.8052 (2025): 1065-1075.
>
> [3] Dong, Mingze, et al. "Causal identification of single-cell experimental perturbation effects with CINEMA-OT." Nature methods 20.11 (2023): 1769-1779.
>
> [4] Adduri, Abhinav K., et al. "Predicting cellular responses to perturbation across diverse contexts with State." bioRxiv(2025): 2025-06.
>
> [5] Klein, Dominik, et al. "CellFlow enables generative single-cell phenotype modeling with flow matching." bioRxiv (2025): 2025-04.
>
> [6] Weinberger, Ethan, Chris Lin, and Su-In Lee. "Isolating salient variations of interest in single-cell data with contrastiveVI." Nature Methods 20.9 (2023): 1336-1345.
>
> [7] FLOW MATCHING FOR GENERATIVE MODELING
>
> [8] One step diffusion via shortcut models
>
>
> # **The remaining responses are provided as follow. We hope our responses clarify and kindly request raising the score!!!**

---

> > ### Author Response · Authors · 2025-11-16
> > **Response to Reviewer DS5U (2/2)**
> >
> > >  Question about Fig.5
> >
> > **Answer:** In Fig. 5, for each perturbation condition, we first collect all cells under that condition. We then compute, for every gene, the fraction of cells in which the gene is expressed, which is used as the ground-truth activation probability.
> >
> > We do not explicitly distinguish biological from technical zeros. This distinction is **fundamentally non-identifiable in scRNA-seq data, and the dataset provides no annotations indicating the origin of each zero.** Our model therefore treats zeros as part of the observed expression distribution, consistent with standard practice.
> >
> > >  Comparison with linear model?
> >
> > **Answer:** We evaluate the linear model on the Adamson dataset. While the **linear model performs competitively and indeed surpasses several deep learning baselines**, it still falls **noticeably short of Doloris across all evaluation metrics.**
> >
> > | Method     | E_distance_all | EMD_all | E_distance_DE20 | EMD_DE20 | E_distance_DE40 | EMD_DE40 |
> > |------------|----------|----------|----------|----------|----------|----------|
> > | linear model      |0.8722±0.0775|0.0336 ±0.0023|0.8967 ±0.1582|0.1784 ±0.0278|0.9414±0.0433|0.1699 ±0.0212|
> > | Doloris    |0.5035 ±0.1698|0.0364 ±0.0038|0.5312 ±0.1937|0.0833 ±0.569|0.5352 ±0.2085|0.0804 ±0.0481|
> >
> > >  More comprehensive metrics
> >
> > **Answer:** We evaluated our method using commonly adopted metrics in Cell-eval metrics [4] in Sci-Plex3, which show strong performance. However, these metrics typically consider only mean values, while cells under the same perturbation often exhibit heterogeneity—some genes even display bimodal expression—so they may not fully capture distributional differences. **Nevertheless, based on our distribution-aware metrics, our model demonstrates consistently robust performance.**
> >
> > | Method     | MAE | RMSE | PDiscNorm (↑) |
> > |------------|----------|----------|----------|
> > | chemCPA | 0.0293 ±0.0021 | 0.0570 ±0.0130 | 0.5476 ±0.1393|
> > | BioLord | 0.0228 ±0.0033 | 0.0409 ±0.0180 |  0.5123 ±0.1028|
> > | Doloris      | 0.0139 ±0.0026 | 0.0287 ±0.0157| 0.6900 ±0.1315|
> >
> > [1] Bunne, Charlotte, et al. "Learning single-cell perturbation responses using neural optimal transport." Nature methods 20.11 (2023): 1759-1768.
> >
> >  [2] Klein, Dominik, et al. "Mapping cells through time and space with moscot." Nature 638.8052 (2025): 1065-1075.
> >
> > [3] Dong, Mingze, et al. "Causal identification of single-cell experimental perturbation effects with CINEMA-OT." Nature methods 20.11 (2023): 1769-1779.
> >
> > [4] Adduri, Abhinav K., et al. "Predicting cellular responses to perturbation across diverse contexts with State." bioRxiv(2025): 2025-06.
> >
> > [5] Klein, Dominik, et al. "CellFlow enables generative single-cell phenotype modeling with flow matching." bioRxiv (2025): 2025-04.
> >
> > [6] Weinberger, Ethan, Chris Lin, and Su-In Lee. "Isolating salient variations of interest in single-cell data with contrastiveVI." Nature Methods 20.9 (2023): 1336-1345.
> >
> > [7] FLOW MATCHING FOR GENERATIVE MODELING
> >
> > [8] One step diffusion via shortcut models
> >
> >
> > # **We hope our responses clarify and kindly request raising the score!!!**

---

> > > ### Comment · Reviewer_DS5U · 2025-11-19
> > >
> > > Thank you for the rebuttal. The added experimental results strengthen the paper and resolve several of my concerns. However I remain hesitant on the following points:
> > > 1. Doesn't ContrastiveVI [6] also avoid explicit pairing? The current rebuttal avoids mentioning this and therefore misleading.
> > > 2. Modeling the dropout is an established practice in scRNA-seq data analysis, which associates with the sequencing process and has (or should have) nothing to do with gene-gene level dependencies. I find this part of the rebuttal also tricky and a bit misleading. It would be a big claim if the dropouts in scRNA-seq data indeed associates with real biology.
> > > 3. The way the rebuttal avoids showing mean metrics is also concerning. How can I trust the distributional behaviors if the method does not output the correct mean? In particular, if a method cannot outperform linear model in terms of mean metrics, then it would have limited practical utility for identifying top gene targets and performing mechanism analysis.
> > >
> > > I am increasing my score to 4, conditioning on that authors acknowledge these and revise the manuscript accordingly.

---

> > > > ### Author Response · Authors · 2025-11-27
> > > > **We have updated the manuscript following your valuable advice**
> > > >
> > > > Thanks very much for your feedback! Your suggestions helped us better position our contributions within the existing literature and avoid potential omissions. Following your valuable advice, we have updated the corresponding part of the manuscript (in Appendix A.6 and A.7) and added citations to the relevant papers [1,2,3] you recommended.
> > > >
> > > > [1] Bunne, Charlotte, et al. "Learning single-cell perturbation responses using neural optimal transport." Nature methods 20.11 (2023): 1759-1768.
> > > >
> > > > [2] Dong, Mingze, et al. "Causal identification of single-cell experimental perturbation effects with CINEMA-OT." Nature methods 20.11 (2023): 1769-1779.
> > > >
> > > > [3] Adduri, Abhinav K., et al. "Predicting cellular responses to perturbation across diverse contexts with State." bioRxiv(2025): 2025-06.

---

> ### Author Response · Authors · 2025-11-20
> **Clarification regarding the misunderstanding between us**
>
> Thanks very much for your professional, insightful, and helpful reviews! We greatly enjoy such in-depth
> discussions and believe they will significantly enhance the quality of our work.
>
> >  "Doesn't ContrastiveVI [6] also avoid explicit pairing?"
>
> **Answer:** While ContrastiveVI does avoid explicit pairing, it is important to emphasize that this is because **its task is reconstruction-based representation learning, not modeling transitions between conditions.**
>
> **It does not model or predict gene expression under new or unseen perturbations, nor does it address the unpaired transition problem that defines single-cell perturbation prediction.**
>
> In our setting, the absence of paired control–perturbation data is a core challenge precisely because **we aim to predict a transition from control cell distribution to perturbed cell distribution, not to perform contrastive analysis.**
>
> > Modeling the dropout with gene-gene dependencies is big claim
>
> **Answer:** We agree that observed zeros reflect technical dropout. **However, numerous studies have shown that dropout is not purely random, and many observed zeros carry meaningful biological information [1,2,3].** Therefore, leveraging gene–gene dependencies to predict gene activation states is fully justified in this setting and does not constitute a “big claim”
>
> > Question about mean metrics
>
> **Answer:** We supplemented the expected metrics on the gene perturbation dataset (Adamson), and our method indeed outperforms the linear model in terms of these metrics.
>
> Our intention is to highlight inherent limitations of relying solely on mean metrics—particularly for genes with heterogeneous or bimodal expression distributions—where the mean may fail to capture biologically relevant variability. We believe that **drawing attention to these limitations encourages the community to critically assess evaluation criteria and promotes more meaningful scientific interpretation.**
>
> | Method     | MAE | RMSE | PCC_delta |
> |------------|----------|----------|----------|
> | linear model      |0.0292±0.0021|0.0544 ±0.0092|0.6367 ±0.0677|
> | Doloris    |0.0214 ±0.0026|0.0366 ±0.0115|0.6898 ±0.0386|
>
> [1] Bayesian model selection reveals biological origins of zero inflation in single-cell transcriptomics
>
> [2] Embracing the dropouts in single-cell rna-seq analysis
>
> [3] Statistics or biology: the zero-inflation controversy about scRNA-seq data

---

### Official Review · Reviewer_hoSz · 2025-10-27

**Soundness:** 3
**Presentation:** 3
**Contribution:** 3
**Rating:** 6
**Confidence:** 5

**Summary:**

This author with this paper introduces DOLORIS, a new generative framework for predicting the gene expression of single cells after perturbation. The work is well-motivated by the unique challenges of single-cell data: it is unpaired, high-dimensional, and sparse. The proposed solution combines a dual diffusion bridge model with a dedicated sparsity masking mechanism.

**Strengths:**

***S1:*** The application of the DDIB framework is an excellent fit for the unpaired nature of the problem. It provides a principled way to bridge the control and perturbed distributions without imposing strong, potentially incorrect, assumptions about cell-to-cell correspondence.

***S2:*** The sparsity masking strategy is a very clever and pragmatic contribution. High-dimensional, sparse data is a major challenge for many generative models, which can easily waste capacity modeling the uninformative zero values. Decoupling the prediction of sparsity from the prediction of expression values is a clean solution that appears to work very well based on the ablation studies.

***S3:*** The work addresses a problem of significant practical importance in drug discovery and functional genomics. Developing more accurate predictive models for perturbation effects has the potential to dramatically accelerate biomedical research. The paper does a good job of motivating the problem and its real-world impact.

**Weaknesses:**

***W1*** While the DDIB framework is a good fit, the paper does not provide a strong argument for why it is superior to other generative frameworks for unpaired domain translation (e.g., those based on VAEs or GANs, like CycleGAN). A brief discussion contextualizing this choice would strengthen the paper.

***W2*** The model consists of two key parts: the mask model and the diffusion model. The paper evaluates them in ablation studies but does not analyze their interaction. For example, how robust is the diffusion model to potential errors made by the mask model? Does a slightly imperfect mask lead to catastrophic failures or graceful degradation in the generated expression?

***W3*** The model is conditioned on specific perturbation information. While it is tested on unseen perturbations, it's not entirely clear how it would generalize to entirely new classes of perturbations not seen during training. The representation of the perturbation itself seems crucial, and this aspect is not deeply explored.

**Questions:**

Beside weakness, I think would be beneficial if authors could briefly answer to some additional questions.

***Regarding the Sparsity Mask:*** Could you elaborate on the potential failure modes of the mask model and how they might impact the final predictions? For instance, if the mask model incorrectly predicts a biologically crucial gene as "silent" (inactive), does your framework have any mechanism to recover, or is that information irrecoverably lost?

***Regarding the DDIB Framework:*** Could you provide more intuition on why the shared latent space is sufficient to learn a biologically meaningful transformation? How does the model ensure that a control cell is mapped to a corresponding perturbed cell, rather than just an arbitrary cell from the perturbed distribution that matches the conditioning information?

***Regarding Computational Cost:*** Your model involves two diffusion models plus a mask model. Could you comment on the computational resources required and the inference time per sample compared to the baseline methods?

---

> ### Author Response · Authors · 2025-11-14
> **Response to Reviewer hoSz**
>
> We would like to thank the reviewer for the positive and insightful assessment of our work. We greatly appreciate the reviewer’s recognition of the conceptual contributions of our method, as well as the thoughtful questions that pinpoint important technical nuances. We provide detailed responses below.
>
> > # Why choose DDIB?
>
> **Answer:** We justify our choice from both theoretical and practical perspectives:
>
> **Theoretically:**
>
> 1. DDIB’s probability flow ODEs solve a special Schrödinger Bridge Problem[1], providing an OT-efficient translation that alternative unpaired methods (e.g., GANs, VAEs) lack.
>
> 2. DDIB is based on PF-ODEs, **so it satisfies cycle consistency almost automatically in both theory and practice[1,3].** In contrast, GAN-based methods such as **CycleGAN require additional loss terms and joint training to approximate this property**, making **DDIB inherently more stable and structurally advantageous.**
>
> **Practically:**
>
> 1. GAN-based methods are prone to **training instability, and mode collapse, making them difficult to train reliably[2].** VAEs, though more stable, **often produce blurred and less diverse samples [4].** In contrast, diffusion models train reliably and generate high-quality, diverse outputs.
>
> 2. Some unapired transition methods (CycleGAN) **require two independent mappings to enforce invertibility**, which involves learning two independent mappings: a forward mapping from source to target and a backward mapping from target to source. In conditional generation, the **backward mapping is often ill-posed[5,6], as it involves decoupling complex target information to recover the original input, which is inherently ambiguous.** In single-cell perturbation, predicting control states from perturbed results **involves counterfactual inference, making it much harder**, and **at least doubles the parameter count.**
>
> > # Regarding the Sparsity Mask
>
> **Answer:** To demonstrate the diffusion model's robustness to potential mask errors, we conducted an experiment using independent Bernoulli (i.i.d.) sampling for each gene. This approach partially respects the predicted probabilities but does not capture dependencies across genes. The results of i.i.d did not fully collapse, **indicating that the diffusion model maintains robustness even under imperfect mask conditions.**
>
> While genes predicted as inactive cannot be recovered, **our mask is designed from the outset to minimize such errors through probabilistic outputs and careful sampling.** For genes predicted as 1 (active), the diffusion model can robustly generate reasonable expression values, even if the mask is slightly imperfect.
>
> | Method     | E_distance_all | EMD_all | E_distance_DE20 | EMD_DE20 | E_distance_DE40 | EMD_DE40 |
> |------------|----------|----------|----------|----------|----------|----------|
> | i.i.d.|0.5351±0.2741|0.0311 ±0.0061|0.3244 ±0.1914|0.0882 ±0.0254|0.3500±0.1880|0.0779 ±0.0256|
> | Base model        |0.4055 ±0.2190|0.0265 ±0.0051|0.2484 ±0.1710|0.0743 ±0.0216|0.2406 ±0.1671|0.0649 ±0.0219|
>
> > # How to represent perturbation information?
>
> **Answer:** For gene perturbations, we follow the approach in [7] using GRN-based GNN module (Appendix A.5). Specifically, for gene knockouts, we mask the node corresponding to the knocked-out gene in the GNN and then aggregate to obtain a perturbation representation (Section 3.7). For molecule perturbations, we first extract molecular embeddings and then inject them into the nodes of the GNN for aggregation, obtaining perturbation representation. **Further implementation details are provided in the supplementary materials, including the code.**
>
> > # Computational Cost
>
> **Answer:** Assuming the forward and reverse processes each have a cost of `N` and the mask model has a lightweight cost of `1`, the theoretical cost per sample is `2N + 1`.
>
> In large-scale perturbation generation, where the number of control cell types is `1` and there are `k` perturbation conditions with `n` samples each, we can **reuse the latent from the forward process** instead of regenerating. Letting `m` be the number of control cells used in the forward process, the total computational cost becomes `(1 + N) * k * n + m * N`.
>
> For moderate `m` and large `k`, the computational cost is comparable to a diffusion model that directly denoises `k * n` samples (`N * k * n`).
>
> [1] DUAL DIFFUSION IMPLICIT BRIDGES FOR IMAGE-TO-IMAGE TRANSLATION
>
> [2] Denoising Diffusion Probabilistic Models
>
> [3] DENOISING DIFFUSION IMPLICIT MODE
>
> [4] Diffusion Models Beat GANs on Image Synthesis
>
> [5] Data-conditional diffusion bridges
>
> [6] Simplified diffusion Schrödinger bridge
>
> [7] GRAPE: Heterogeneous Graph Representation Learning for Genetic Perturbation with Coding and Non-Coding Biotype
>
> # **Your recognition of our approach is especially meaningful, as your support would be invaluable in highlighting the contributions of our work to the community. We hope our responses clarify and kindly request raising the score!!!**

---

> ### Comment · Reviewer_hoSz · 2025-11-18
>
> Thanks for addressing all my points, especially the final questions rather than weaknesses. Indeed, it’s thanks to yours explanation to these key points that I’ve decided to raise my score. Nice done.
> **Just one side comment: you don’t need to ask for any raise of the score. It’s not a battle here, it’s peer reviewing.**

---

> > ### Author Response · Authors · 2025-11-19
> > **Appreciation for Your Reconsideration**
> >
> > Thank you very much for taking the time to reevaluate our work and for increasing the score. We sincerely appreciate your thoughtful consideration and the constructive feedback, which helped us improve the paper.

---

### Official Review · Reviewer_NHyz · 2025-11-01

**Soundness:** 2
**Presentation:** 2
**Contribution:** 2
**Rating:** 4
**Confidence:** 3

**Summary:**

The paper introduces Doloris, a novel generative framework for predicting the gene expression of single cells after a perturbation. The paper addresses two main challenges: (1) the data is unpaired, as one cannot measure the same cell before and after perturbation, and (2) gene expression data is extremely sparse. Doloris addresses the unpaired problem by training two separate diffusion models and aligning them through a shared latent space (without explicit pairing). To handle sparsity, the paper proposes a two-part model, namely a mask model that predicts a binary mask of which genes will be silent, and a diffusion model trained only on the active (non-zero) genes.

**Strengths:**

The dual-bridge framework is a novel and effective solution to the fundamental unpaired data problem.

The paper's emphasis on using distributional metrics (E-distance, EMD) over simple RMSE is well-argued, as these metrics are better suited to capturing the cellular heterogeneity inherent in single-cell data.

**Weaknesses:**

A significant weakness is the lack of details for the core diffusion models. The paper does not state what neural network architecture is used as the backbone.

The authors state they use a GAT to embed perturbation information. However, this choice seems arbitrary. The paper provides no ablation studies or comparisons to other GNN architectures.

The paper fails to connect its sparsity masking strategy to existing, well-established statistical frameworks. This two-part architecture with one model for the zero/non-zero binary outcome and a second model for the non-zero continuous values, is a direct implementation of a Hurdle Model. Drawing this parallel would have strengthened the paper by grounding its components in classical statistical theory.

**Questions:**

What is the specific architecture of the diffusion models?

What is the specific architecture of the mask model? Is it also conditioned by the GAT embedding?

I think the paper proposes a complex sampling strategy for the binary mask to ensure global consistency. How does this compare to simply thresholding the mask model's probabilities at 0.5? What was the performance gain from this more complex method?

---

> ### Author Response · Authors · 2025-11-13
> **Response to Reviewer NHyz**
>
> Thanks for your careful reviews!
>
> > Lack of details for the core diffusion model
>
> **Answer:** Thanks for your careful reviews! The core diffusion architecture consists of multiple fully connected (MLP) layers interleaved with graph neural network (GNN) modules, which serve as conditional representation components to embed perturbation information and propagate regulatory signals across genes. **The full architectural details and implementation code are provided in the Supplementary Materials.**
>
> > Lack of details for the mask model
>
> **Answer:** We appreciate the reviewer’s comment. The mask model is an independent regression network composed GNN layers. It learns from the same GRN prior but is not conditioned on the GAT embeddings from the diffusion model. The two models are trained separately. **The full architectural details are also provided in the Supplementary Materials.**
>
> > Why GAT？
>
> **Answer:** Thanks for your helpful reviews! We chose GAT because **its attention mechanism allows adaptive weighting of gene–gene interactions**, which is particularly useful for modeling regulatory effects under perturbations. Preliminary tests with GCN showed slightly lower performance (ablation on sciplex3 dataset, table below), but **we emphasize that the choice of GAT is primarily motivated by its ability to capture dynamic regulatory relationships.**
>
> | Method     | E_distance_all | EMD_all | E_distance_DE20 | EMD_DE20 | E_distance_DE40 | EMD_DE40 |
> |------------|----------|----------|----------|----------|----------|----------|
> | GCN        |0.4237 ±0.1997|0.0271 ±0.0050|0.2645 ±0.1683|0.0774 ±0.0203|0.2809 ±0.1488|0.0691 ±0.0114|
> | Base model (GAT)        |0.4055 ±0.2190|0.0265 ±0.0051|0.2484 ±0.1710|0.0743 ±0.0216|0.2406 ±0.1671|0.0649 ±0.0219|
>
> > Connection with classical Hurdle Models
>
> **Answer:** We thank the reviewer for this insightful and thoughtful comment. **The connection to classical Hurdle Models is indeed very relevant, and your suggestion highlights an important and enlightening statistical perspective.** In future revisions, **we will explicitly mention and discuss the link to Hurdle Models to better ground our sparsity masking strategy in established statistical theory.**
>
> > Why complex sampling strategy?
>
> **Answer:** Thank you for this valuable feedback! As the table shows, using a fixed threshold performs poorly. This is because **it ignores the probabilistic information from the model, and in high-dimensional gene space, small per-gene errors accumulate across thousands of genes.** As a comparison, we also considered independent Bernoulli sampling for each gene (i.i.d.), which partially accounts for probabilities but still fails to capture dependencies across genes. **Our complex sampling strategy preserves the learned probabilities and ensures global consistency, leading to more coherent and reliable predictions.**
>
> | Method     | E_distance_all | EMD_all | E_distance_DE20 | EMD_DE20 | E_distance_DE40 | EMD_DE40 |
> |------------|----------|----------|----------|----------|----------|----------|
> | threshold 0.5        |2.2100 ±0.2252|0.0491 ±0.0059|0.8908 ±0.2544|0.1249 ±0.0690|1.0725 ±0.1964|0.1142 ±0.0504|
> | i.i.d.|0.5351±0.2741|0.0311 ±0.0061|0.3244 ±0.1914|0.0882 ±0.0254|0.3500±0.1880|0.0779 ±0.0256|
> | Base model        |0.4055 ±0.2190|0.0265 ±0.0051|0.2484 ±0.1710|0.0743 ±0.0216|0.2406 ±0.1671|0.0649 ±0.0219|
>
>
> # **We hope our responses clarify and kindly request raising the score!!!**

---

> ### Author Response · Authors · 2025-11-27
> **We have updated the manuscript following your valuable advice**
>
> We thank the reviewer for raising this insightful point regarding our use of a GAT for perturbation embedding. In response to this comment, we have added further explanation and clarification in Appendix A.5, where we discuss the rationale behind this architectural choice. We sincerely appreciate the reviewer’s valuable suggestion !

---

### Official Review · Reviewer_r3Gg · 2025-11-02

**Soundness:** 2
**Presentation:** 2
**Contribution:** 2
**Rating:** 4
**Confidence:** 3

**Summary:**

The authors tackle the problem of estimating single-cell gene expression under genetic or molecular perturbations, where data is inherently unpaired due to the destructive nature of sequencing. They introduce Doloris, which uses two conditional diffusion models: a "source" model for unperturbed (control) cells and a "target" model for perturbed cells, implicitly aligned via a shared Gaussian latent space using Dual Diffusion Implicit Bridges (DDIB). To handle sparsity (abundant zeros in expression data), they add a mask model that predicts silent genes, allowing the diffusion process to focus on expressed patterns. During inference, continuous expressions are generated via DDIB, then masked and rescaled. The framework conditions on cell types and perturbations.

**Strengths:**

Single-cell perturbation modeling is difficult because scRNA-seq is destructive. We can’t see “before and after” for the same cell. Data is unpaired and high-dimensional + sparse. Existing models often (i) implicitly force pairing or (ii) ignore the unpaired nature, or (iii) regress to the mean and miss heterogeneity. Doloris learns a source conditional diffusion model for control cells and a target conditional diffusion model for perturbed cells, make them share a latent Gaussian space via ODE mapping, so a real control cell is diffused to latent, then denoised to the perturbed state under a specified perturbation. Additionally, they introduce a mask model that predicts which genes should be silent for a given perturbation and cell-type condition. Losses for diffusion are computed only on expressed genes.

**Weaknesses:**

1. The abstract boasts SOTA on public datasets, but misses comparison against recent unpaired optimal transport methods like scOT, diffusion-based OT or flow matching based OT.
2. The shared latent space assumes control and perturbed distributions are bridgeable via Gaussian priors, but how is this validated? In Sec. 3.4, adding noise to unperturbed means to preserve heterogeneity is ad-hoc. Why Gaussian, and what's the sensitivity to σ_ct? If perturbations drastically shift distributions, this might fail.
3. While the method predicts realistic expression distributions, the paper doesn’t show many biological case studies. For example, does the model recover known pathway-level responses?

**Questions:**

1. In Fig. 2, why those specific genes (AARS, CARS, etc.)? Are they representative, or cherry-picked? Extend to full dataset stats?
2. How robust is the model to perturbation types? For example, does it handle dosage dependent molecular perturbations, or only binary knockouts?
3. Do you have any downstream validation? Beyond metrics, does it recover known biology, like perturbed pathways in KEGG?

---

> ### Author Response · Authors · 2025-11-14
> **Response to Reviewer r3Gg (1/2)**
>
> Thanks for your careful reviews!
>
> > Miss comparison against recent unpaired optimal transport methods
>
> **Answer:** Thank you for the suggestion. We were **unable to locate any published methods named “scOT,” “diffusion-based OT,” or “flow matching based OT,” nor did we find evidence of their application or empirical results on single-cell perturbation prediction tasks**.
>
> If you could kindly **provide the specific references (paper title / authors / DOI or arXiv link)**, we would be happy to include the corresponding comparison and discussion in the revised manuscript.
>
> Besides, **we would like to emphasize that our work is fundamentally application-driven, rather than proposing a new unpaired matching algorithm. A key contribution of this study is the adaptation of the DDIB framework to the single-cell perturbation prediction setting, which requires nontrivial domain-specific architectural and modeling decisions.** Moreover, unpaired transition methods may not directly transferable to this task and typically require substantial task-specific modifications or methodological extensions before they can operate effectively.
>
> > Question about shared latent space
>
> **Answer:** In diffusion, the **data distribution is effectively aligned to a standard Gaussian distribution** in the latent space[1,2]. In our work, the source model learns a diffusion model over control cells, while the target model learns a diffusion model over perturbed cells. **Both models align their respective data distributions to the standard Gaussian latent space.** As a result, **the source and target models naturally share a latent space by construction**, which does not require any additional assumption about the distributions being “bridgeable” in the original data space[3].
>
> > Question about Sec 3.4 (Eq.12)
>
> **Answer:** We would like to clarify that **the control-group information in Eq.12 is used solely as an additional conditional input to the target diffusion model.** The target model **still learns a standard diffusion mapping that aligns the perturbed cells with a Gaussian latent space.** The control information **does not force any alignment between the control and perturbed distributions; it merely informs the model of the pre-perturbation state.** Therefore, concerns about the method failing if the perturbed distributions are drastically shifted are unfounded.
>
> In the target diffusion model, **we aim to provide the model with information about the control group**. Directly **using real control samples would implicitly create artificial pairings with unpaired perturbations**. Instead of assuming a specific distribution, we **inject gene-wise noise, scaled by the control-group variance**, around the mean expression. This approach preserves the overall expression structure while effectively modeling the heterogeneity among control cells.
>
> > Question about Fig.2
>
> **Answer:** We would like to clarify that the genes shown in Fig. 2 correspond to the perturbed genes in the testing set.
>
> Below we report the intra-condition distances for all perturbations (Adamson) test set. The full results show that we consistently preserves realistic variability and avoids collapse, confirming that the examples in Fig. 2 are representative of the overall performance.
>
> | Pert_cond\ intra distance    | GT   | Doloris | scLambda |
> |--------------|----------|----------|----------|
> | HSD17B12     | 20.435   |21.223 | 12.482 |
> | P4HB         | 19.658   | 19.301 |13.937|
> | AARS         | 22.047   |21.646 | 11.073|
> | DAD1         | 19.743   | 20.231 |13.521|
> | CHERP        | 18.614   |19.341 |11.882|
> | DHDDS        | 20.950   |19.741|13.104|
> | TMEM167A     | 19.848   | 21.268 |13.693|
> | SPCS2        | 19.458   |18.721 |14.357|
> | SLC35B1      | 19.520   |20.674 |15.186|
> | TIMM23       | 19.814   | 21.371|12.902|
> | CARS         | 23.126   |21.166|13.802 |
> | CAD          | 19.832   | 19.540|14.921|
> | HYOU1        | 19.465   | 20.214|11.007|
> | SRPR         | 19.699   |18.439 |12.266|
> | SEC63        | 19.345   |20.436|12.742|
> | PDIA6        | 19.351   | 21.323 |13.988|
> | TTI2         | 19.198   |  20.071 |13.447|
> | PTDSS1       | 19.446   | 20.872 |14.604|
> | COPZ1        | 21.175   | 20.933 |12.697|
> | PPWD1        | 20.968   | 20.893 | 13.319 |
> | EIF2B2       | 21.259   | 21.066 | 13.701 |
> | STT3A        | 19.565   | 18.479  |12.885|
>
>
> [1] Understanding Diffusion Models: A Unified Perspective
>
> [2] Denoising diffusion probabilistic models
>
> [3] DUAL DIFFUSION IMPLICIT BRIDGES FOR IMAGE-TO-IMAGE TRANSLATION
>
>
> # **The remaining responses are provided in the following chat box. We hope our responses clarify and kindly request raising the score!!!**

---

> > ### Author Response · Authors · 2025-11-14
> > **Response to Reviewer r3Gg (2/2)**
> >
> > > Support dose-dependent drug perturbations?
> >
> > **Answer:** Our model is not limited to binary gene knockouts. For molecular perturbations, we **explicitly encode both the drug identity and the dosage level.** This allows the model to **capture dosage-dependent responses rather than treating perturbations as binary events.**
> >
> > **The detailed implementation for handling dosage-dependent molecular perturbations is provided in the supplementary code.**
> >
> > > Recover known biology?
> >
> > **Answer:** For the predicted perturbation results under each perturbation condition, we selected the top 100 genes based on logFC and performed KEGG pathway enrichment analysis. The top 3 most significantly enriched pathways are shown below:
> >
> > **gene perturbation HYOU1**
> > | Term                                        | Adjusted P-value | Genes                                                                                                                   |
> > | ------------------------------------------- | ---------------- | ----------------------------------------------------------------------------------------------------------------------- |
> > | Ribosome                                    | 5.01e-21         | RPL10;RPL32;RPS8;RPL12;RPL34;RPS6;RPL13A;RPS3A;RPL7;RPS27;RPS16;RPS15A;RPS18;RPL14;RPL26;RPS27A;RPL28;RPS24;RPS23;RPS12 |
> > | Pathogenic Escherichia coli infection       | 6.32e-07         | TUBA1B;ARPC1B;TUBB;NCL;ARPC5;ACTB;ACTG1                                                                                 |
> > | Protein processing in endoplasmic reticulum | 5.70e-06         | PDIA3;HSP90AB1;HSPA5;SSR4;RPN2;DAD1;ERP29;PDIA6;ATF4                                                                    |
> >
> > **gene perturbation PPWD1**
> > | Term | Adjusted P-value | Genes |
> > |------|----------------|-------|
> > | Ribosome | 3.40e-24 | RPL10;RPL32;RPS8;RPL12;RPL34;RPS6;MRPL16;RPL13A;RPS3A;RPL7;RPS26;RPS16;RPS15A;RPS18;RPL37A;RPL14;RPL38;RPS27A;RPL28;RPS10;RPS23;RPS12 |
> > | Protein processing in endoplasmic reticulum | 1.29e-10 | PDIA3;HSPA8;HSP90AA1;HSP90AB1;HSPA5;RPN2;PDIA6;HSP90B1;LMAN2;CANX;ERP29;CALR;SEC61B |
> > | Antigen processing and presentation | 7.30e-09 | PDIA3;HSPA8;HSP90AA1;HSP90AB1;HSPA5;CTSL;CANX;CALR;B2M |
> >
> >
> > The very low adjusted P-values indicate that the predicted top genes are significantly concentrated in these pathways. This suggests that the predicted perturbation captures relevant biological processes, including protein synthesis (Ribosome), cellular stress response (Pathogenic E. coli infection), and protein folding/quality control in the ER (Protein processing in endoplasmic reticulum). Overall, this downstream analysis demonstrates that our model not only achieves numerical accuracy but also recovers known biology.
> >
> > # **We hope our responses clarify and kindly request raising the score!!!**

---

### Author Response · Authors · 2025-11-29
**A brief summary of the key pros and cons for the convenience of (senior) area chairs**

Dear (senior) ACs,

We extend our sincere gratitude for the time and effort you have dedicated to review our manuscript.

This paper introduces **a novel paradigm for single-cell perturbation modeling, addressing unpaired data alignment and using a sparsity-masking strategy to mitigate mode collapse in high-dimensional gene expression.** Addressing unpaired data remains one of the biggest bottlenecks in single-cell perturbation modeling today, and **our work offers a promising approach to overcome this fundamental challenge, while also mitigating issues arising from the sparsity of high-dimensional gene expression data.** We hope to encourage the AI4Science community to focus more attention on this critical problem to advance the field.

> # Key pros noted by the reviewers:

**S1.** Motivation or research problem is good/interesting. (3/4 Reviewers r3Gg, NHyz, hoSz)

**S2.** Idea is novel/interesting/reasonable. (2/4 Reviewers NHyz, hoSz)

**S3.** The experimental are comprehensive/ well-designed/ promising. (3/4 Reviewers r3Gg, NHyz, hoSz)

**S4.** The paper’s presentation is generally clear. (4/4 Reviewers r3Gg, NHyz, hoSz, DS5U)

> # Outstanding reviewer issues
---
# **Issues of Reviewer r3Gg**
**Resolved/Answered issues:**

✅ 1. We clarified that the source and target diffusion models naturally share a Gaussian latent space, which does not require any additional assumption about the distributions being “bridgeable” in the original data space.

✅ 2. We explained that control-group information is used only as a conditional input and does not enforce artificial alignments, while gene-wise noise preserves heterogeneity, not ad-hoc.

✅ 3. We verified that predicted perturbation responses recover known biology through KEGG pathway enrichment and provided detailed explanations addressing the reviewer’s questions on the figure 2.

✅ 4. Our model accurately handles dosage-dependent perturbations by explicitly encoding both drug identity and dosage levels.

**Unresolved issues:**

❌ 1. Reviewer r3Gg suggested comparison to unpaired transition methods, **but relevant baselines are either unrelated or untraceable, and the reviewer provided no further guidance.**

---

# **Issues of Reviewer NHyz**
**Resolved/Answered issues:**

✅ 1. Our core diffusion architecture, combining MLP and GNN components, is fully documented in the Supplementary Materials.

✅ 2. We clarified the design of the mask model as an independent GNN-based network trained separately from the diffusion model.

✅ 3. We justified the use of GAT for adaptive gene–gene interactions and provided ablation results showing slightly lower performance with GCN.

✅ 4. The sparsity masking mechanism aligns with Hurdle Models, and this connection has been explicitly clarified in the updated appendix.

✅ 5. We demonstrated that our complex sampling strategy preserves global gene dependencies and produces coherent predictions, unlike thresholding or i.i.d. sampling.

**Unresolved issues:**

No. **Reviewer did not provide any follow-up feedback.**

---

# **Issues of Reviewer hoSz**

**Resolved/Answered issues:**

✅ 1. We justified the choice of DDIB, highlighting its theoretical advantages (solving a Schrödinger Bridge problem with cycle consistency) and practical benefits (stable training, high-quality and diverse outputs) over GANs and VAEs, as well as its suitability for counterfactual inference in single-cell perturbation.

✅ 2. In rebuttal, we demonstrated the diffusion model’s robustness to mask errors via i.i.d. sampling experiments, showing that the model maintains reliable predictions even with imperfect masks.

✅ 3. We explained how perturbation information is represented for both gene knockouts (via masked GNN nodes) and molecular perturbations (via molecular embeddings injected into GNN nodes).

✅ 4. We provided a computational cost analysis, showing that generating multiple perturbation conditions is efficient and comparable to standard diffusion denoising approaches.

**Unresolved issues:**

No. **All raised points have been addressed and clarified to the reviewer’s satisfaction, resulting in a score of 8.**

# **Issues of Reviewer DS5U**

**Resolved/Answered issues:**

✅ 1. We clarified that Doloris predicts post-perturbation distributions from control cells and perturbations, unlike most cited ‘unpaired’ methods that rely on known source and target cells or focus on different tasks.

✅ 2. We demonstrated the necessity of multi-step generation for high-dimensional single-cell data: one-step generation severely degrades reconstruction quality.

✅ 3. Results in rebuttal shows that **Doloris outperforms linear model on both mean-based and distribution-based metrics.**

**Unresolved issues:**

❌ 1. Consensus was not reached on our use of gene–gene dependencies (cell-level) to predict activation from observed zeros, though we provided references showing many zeros carry meaningful biological information.

---

### Meta-Review · Area_Chair_WNWb · 2026-01-04

**Summary:**

This paper develops a generative framework for for single-cell perturbation modeling that address the issues of unpaired and sparse data.

Reviewers generally found the motivation strong and the proposed approach novel/practical. While initial concerns were pointed out regarding e.g. architectural details, comparisons to linear baselines, and biological validation, the authors' rebuttals have addressed most of these concerns.

**Reviewer Concerns:**

Concerns addressed by rebuttals:
- Reviewer NHyz asked questions regarding model architecture, such as the core diffusion models, mask model, and use of GAT, which the authors addressed by providing detailed clarificaitons.
- Reviewer NHyz pointed out connection with the classical Hurdle Models, which the authors addressed by promising to discuss the connection in the revision.
- Reviewer DS5U suggested comparison with linear baselines, which the authors addressed by providing further comparison with them.
- Reviewer DS5U  pointed out the lack of an ablation study on the value of multi-step inference versus a one-step approach, which the authors addressed by providing ablation study.
- Reviewer r3Gg asked whether there is downstream biological validation, which the authors addressed with further empirical analysis.

Outstanding concerns:
- Reviewer r3Gg suggested comparison to recent unpaired optimal transport methods (but the authors were not able to find relevant ones).
- As pointed out by Reviewer DS5U, modeling dropout is an established practice in analysis scNRA-seq and may have nothing to do with gene-gene level dependencies. The authors provide several references to argue that many observed zeros carry meaningful biological information, but a consensus was not reached.

**Reviewer Scores:**

Reviewer hoSz and DS5U decided to increase their score (to 8 and 4, respectively), although they are not reflected on the reviews. Reviewers r3Gg and NHyz may have increased their scores (to 6) if they had been able to participate fully in the discussion.

---

### Decision · Program_Chairs · 2026-01-26

Accept (Poster)